# Role of Divalent Cations in Infections in Host–Pathogen Interaction

**DOI:** 10.3390/ijms25189775

**Published:** 2024-09-10

**Authors:** John A. D’Elia, Larry A. Weinrauch

**Affiliations:** Kidney and Hypertension Section, E P Joslin Research Laboratory, Joslin Diabetes Center, Department of Medicine, Beth Israel Deaconess Medical Center, Harvard Medical School, Boston, MA 02215, USA

**Keywords:** tuberculosis, divalent cations, calcium ion channels, antimicrobial defense, microbial defense, autophagy, apoptosis, pathogen-host interaction

## Abstract

With increasing numbers of patients worldwide diagnosed with diabetes mellitus, renal disease, and iatrogenic immune deficiencies, an increased understanding of the role of electrolyte interactions in mitigating pathogen virulence is necessary. The levels of divalent cations affect host susceptibility and pathogen survival in persons with relative immune insufficiency. For instance, when host cellular levels of calcium are high compared to magnesium, this relationship contributes to insulin resistance and triples the risk of clinical tuberculosis. The movement of divalent cations within intracellular spaces contributes to the host defense, causing apoptosis or autophagy of the pathogen. The control of divalent cation flow is dependent in part upon the mammalian natural resistance-associated macrophage protein (NRAMP) in the host. Survival of pathogens such as *M tuberculosis* within the bronchoalveolar macrophage is also dependent upon NRAMP. Pathogens evolve mutations to control the movement of calcium through external and internal channels. The host NRAMP as a metal transporter competes for divalent cations with the pathogen NRAMP in *M tuberculosis* (whether in latent, dormant, or active phase). This review paper summarizes mechanisms of pathogen offense and patient defense using inflow and efflux through divalent cation channels under the influence of parathyroid hormone vitamin D and calcitonin.

## 1. Introduction

Public health struggles relating to bacterial, viral, or parasitic infection/infestation involve a large percentage of the world population. The expenditures to eradicate debilitating human illnesses brought about by the infections and infestations are extraordinary. Although partial eradication of parasites and their vectors that contribute to human illness may be accomplished, changing climate, mutating organisms, migrating human hosts, famine, and war continue to lead to the need for additional treatments. To develop future therapy, we need a better understanding of how to prevent parasites from piercing the human integument and entering an environment in which they may thrive at the expense of the human organism. *Mycobacterium tuberculosis* is generally airborne and enters through the pulmonary alveoli. Malarial, trypanosomal, and leishmanial forms enter via an insect bite, schistosomes penetrate the skin of the foot in fresh water, and trichinella or toxoplasma are ingested. Whatever the mode of entry, the battle between parasite and host dominance takes place at the cellular and intracellular levels. 

The movement of divalent cations within intracellular spaces contributes to the host defense, causing apoptosis or autophagy of the pathogen. The control of divalent cation flow is dependent in part upon the mammalian natural resistance-associated macrophage protein (NRAMP) in the host [1]. Survival of pathogens such as *M tuberculosis* within the bronchoalveolar macrophage is also dependent upon NRAMP. Pathogens evolve mutations to control the movement of calcium through external and internal channels. The host NRAMP as a metal transporter competes for divalent cations with the pathogen NRAMP in *M tuberculosis* (whether in latent, dormant, or active phase) [2]. Divalent cations are the essential coordinators of cellular and subcellular processes, such as gene expression, cell growth, metabolism, structural coordination, and enzymatic function [3,4,5]. In the following sections, we focus on the role of divalent cations in host–pathogen interactions with particular attention to *M tuberculosis*.

The survival of *M tuberculosis* within the macrophage is dependent upon elements of a cellular host–pathogen relationship. Elevated host calcium levels may be associated with cell injury and the death of the pathogen (apoptosis). With lower host levels of calcium, *M tuberculosis* may survive by repair and recycling by a process known as autophagy. Survival mutations used by Mycobacteria include increased efflux of calcium, with a lesser need for autophagy-related vesicles (phagosomes) [6]. The energy from pumps from the family of P-type ATPase drives calcium out of Mycobacteria [7]. Calcium channel blockers may prevent the efflux of calcium from the mycobacterial pathogen. Calcium channel blockers are useful in stopping the efflux of antibiotics from the pathogen during active infection [8,9]. Calcium and magnesium have similar properties, but in some instances, calcium’s role has been found to be highly specific [8,9]. Patients with an increased ratio of host intracellular calcium to magnesium have an elevated risk of insulin resistance/diabetes mellitus [10]. Diabetes mellitus is associated with a three-fold risk of contracting tuberculosis [11]. 

The storage of calcium in the host endoplasmic reticulum depends upon the interaction between vitamin D, parathyroid hormone, calcitonin hormone, and calmodulin (calcium message transporter). Biologically important divalent ions include calcium (Ca^2+^), cadmium (Cd^2+^), copper (Cu^2+^), iron (Fe^2+^), magnesium (Mg^2+^), manganese (Mn^2+^), selenium (Se^2+^), and zinc (Zn^2+^). Table 1a lists the molecular actions of normal concentrations of Ca^2+.^ Table 1b lists the organ system pathology of toxic concentrations of the divalent cations. Mechanisms requiring anions to couple with divalent cations include calcium (Ca^2+^) and manganese (Mn^2+^) with hydroxyl (-OH) and carboxyl (C=O) groups, while copper (Cu^2+^) and cadmium (Cd^2+^) bind with sulfhydryl (-SH) groups. 

Two mechanisms permitting *M tuberculosis* to remain alive inside of macrophages are the generation of genes controlling the intake/efflux of divalent cations from within the pathogen (3–5) as well as the ability to shut down metabolism and replication into a state of dormancy, by which it can avoid detection by defense mechanisms [12].

## 2. Hypercalcemia in Clinical Tuberculosis

In one well-studied case report, an outdoor worker in India was admitted to the hospital with new-onset pulmonary tuberculosis. The patient had decreased renal function (serum creatinine = 2.6 mg/dL, estimated glomerular filtration rate ~30 mL/minute; ratio of blood urea nitrogen to serum creatinine elevated (69 mg/dL/serum creatinine 2.6 mg/dL = 26) suggesting dehydration, serum calcium elevated at 14.4 mg/dL (upper limit of normal = 10.4), serum phosphate = 2.5 mg/dL (lower limit of normal = 2.5), vitamin D (1,25 dihydroxy-cholecalciferol) level above normal range, alkaline phosphatase normal, parathyroid hormone level low-normal at 11.7 pg/mL (normal range 8.0–74 pg/mL)). Potential mechanisms for the elevation of serum calcium included dehydration and generation or hyperstimulation of vitamin D through the skin, kidney, and bronchoalveolar epithelium [13]. 

Clinical hypercalcemia has been observed in patients with new-onset pulmonary tuberculosis. A potential mechanism involves the synthesis of active vitamin D through the action of pulmonary 1-alpha hydroxylase, which converts the 25 (OH) form of cholecalciferol from liver to 1,25 dihydroxy-cholecalciferol, bypassing the usual renal conversion process. Active vitamin D may be expected to increase the intestinal absorption of calcium. The risk for hypercalcemia is greater when thiazide diuretics are in use, since there is no loss of calcium from the distal renal tubule sodium/potassium exchange site. The resolution of higher concentrations of serum calcium may be expected with the completion of a successful course of antibiotics and corticosteroids [14]. With a bronchial lavage, even people with normal levels of serum calcium while under treatment for pulmonary tuberculosis may demonstrate the active production of 1,25 dihydroxy-cholecalciferol from alveolar macrophages and T-lymphocytes [15]. The same process has been described in the pulmonary macrophages of individuals with sarcoidosis, another pulmonary granulomatous disorder [16]. Hypocalcemia during untreated pulmonary tuberculosis has also been described. The active site may be the bronchial K cell, which is able to secrete calcitonin into the circulatory system. A decreased serum calcium level is to be expected if an elevated serum calcitonin inhibits the action of osteoclasts, which remove calcium phosphorus from bones [17]. Table 2 summarizes the hormones that have important effects on serum calcium. Vitamin D, which is synthesized in the liver/kidney as 1,25 dihydroxy-cholecalciferol, but which has bones as an important target goal, can be considered a vitamin that happens to be a hormone as well. The intracellular transmission of the calcium signal is enhanced by its connection with calmodulin. In the inflammation/immunity functions area, there are increased expressions of intranuclear factors [18] such as that of nuclear factor kappa beta. In the cardiac contraction/relaxation with controlled rhythm function area, the calcium signal is enhanced by its calmodulin connection involving excitation–contraction [19,20].

Many instances of hypercalcemia with untreated pulmonary or disseminated (miliary) tuberculosis are reported from climate zones that are warmer, with a longer exposure to sun, increased skin pigmentation, and a greater likelihood of the hypergeneration of vitamin D. Of the countries reporting hypercalcemia before the treatment of new-onset pulmonary or disseminated tuberculosis (Table 3), only Sweden is in a cooler climate zone [21]. The countries in warmer climate zones include Brazil [13], China [22], Greece [23], India [24], Malaysia [25], Nigeria [26], Pakistan [27], and four warmer locations of the United States of America (USA). These locations include Alabama [28], Kentucky [29], Oklahoma [30], and Tennessee [31]. After many years of decreasing incidence of tuberculosis in the USA, an increased incidence was recorded during the COVID pandemic [32] There was a 35% increase in incidence from 2020 (7171 patients per year) to 2023 (9615 patients per year), during which time the earth experienced its warmest summers and winters [33]. 

## 3. Calcium and Magnesium Deficiency in Pulmonary Tuberculosis with Multiple Cavities in Persons with or without Diabetes Mellitus

European incidence of pulmonary tuberculosis is currently highest in Romania, where multiple cavities may be discovered during the initial examination [33]. The degree of lung pathology is correlated with having levels of serum calcium and serum magnesium below the normal range [33]. In addition, persons with type 2 diabetes mellitus may be deficient in magnesium [10]. The level of blood glucose above the normal range is inversely related to the degree of deficiency of serum magnesium below the normal range [10]. It stands to reason that replacement of magnesium until levels are normal would be associated with an improvement in glucose control [10]. 

## 4. Survival Mechanisms of Mycobacteria Attacked by Macrophages Involve Calcium Extrusion

High levels of calcium are used by mammals to kill single-cell pathogens like *M tuberculosis* [34]. This cation works whenever it surrounds the pathogen, i.e., when it is unprotected in the lung matrix, where apoptosis occurs rapidly, or relatively protected within an alveolar macrophage. Eventually, the macrophage may digest the pathogen invader with or without the help of calcium ions provided from the host. In some instances, the host/macrophage defense may cause a release of calcium from stores within the pathogen itself, causing immediate apoptosis. A combination of fatal and non-fatal events will result in an attempt by the surviving pathogens to clear the area of cell debris (autophagy). The extrusion of calcium from within the Mycobacterium protects it from rapid apoptosis, but not without some injury. The process of self-repair by the injured pathogen can be carried out while it is living dependently within an alveolar macrophage. The intermediate stage of autophagy involves phagosomes. To keep the process going, there must be a process of continuous extrusion of calcium from the pathogen [6,7]. Genetic controls of toxic levels of calcium are in place for the extrusion of other divalent cations (Cu^2+^, Mg^2+,^ Mn^2+^, Zn^2+^) [35,36,37,38]. The extrusion of calcium from within the pathogen protects it from apoptosis, followed by a process of repair. The repair of mycobacteria held inside of macrophages involves the elimination of injured structures by a process known as autophagy. The early stages of autophagy include the encapsulation of each mycobacterium within a structure called a phagosome. In later stages of autophagy, mature phagosomes merge with different structures called lysosomes. To keep the process of autophagy going, there must be a continuous process of extrusion of calcium from the pathogen [6,7]. 

The extrusion of cations across membranes often requires energy through the action of ATPases on ATP to release high-energy phosphate [6,7]. The best known continuously active calcium ion pump system is found in muscles that must contract and relax [sarcolemma endoplasmic reticulum pump (SERCA), see Figure 1]. In mycobacteria, the divalent cation transporting pump system for efflux is genetically controlled through a structure referred to as CptE [6,7]. Autophagy of the pathogen (*M tuberculosis*) is regulated by a process dependent upon the mammalian target of rapamycin (mTOR) [6]. A calcium transporting pump system referred to as CptE in some cells initiates an inflammation cascade involving calmodulin, calcineurin, and nuclear factor kappa beta (nfKB, see Figure 2). Calcineurin inhibitors (cyclosporine, tacrolimus) and calcium channel blockers (amlodipine, nifedipine, diltiazem, verapamil) inhibit this inflammation cascade. Calcineurin, a phosphatase, has a unique anti-inflammatory role to inhibit the merging of phagosomes with lysosomes by means of which the pathogen does not succumb to the later stages of autophagy inside the macrophage [39]. 

Hypercalcemia is often noted in people with pulmonary tuberculosis. The main cause of hypercalcemia is increased serum concentrations of vitamin D. Other causes are listed in Table 3. Two hormones that affect the release of calcium from bones are parathyroid hormone (stimulating osteoclasts) and calcitonin (inhibiting the binding of parathyroid hormone to the osteoclast). Pro-calcitonin appears to be expressed during inflammation, marked by an increase in C-reactive protein concentration [40,41].

## 5. Copper (Cu^2+^) in Elevated Concentrations Can Be Toxic for *M Tuberculosis* and Kidney Function in Persons with Diabetes Mellitus

Copper is absorbed through the gastrointestinal system and transported to the liver bound to albumin. It can be stored in the liver bound to metallothionein [42]. Its transport to the periphery requires a different binding protein called ceruloplasmin [42]. Its excretion is mainly into the bile. Certain structures whose functions depend upon Cu^2+^ include Leiden factor V (natural anti-coagulant), cytochrome C oxidase (electron transport system), lysyl oxidase (bone formation), and dopamine monooxygenase (neurotransmitters) [42]. Increased levels of serum and intracellular copper are associated with an increased incidence of both diabetes mellitus [43,44] and a loss of kidney function through fibrosis [38]. The copper-based enzyme system (lysyl oxidase) has an intracellular connection to renal matrix fibrosis by collagen cross-linking [45].

Copper ion (Cu^2+^) is a reduction/oxidation (redox) agent capable of catalyzing one molecule of hydrogen peroxide (H_2_O_2_) and producing two molecules of hydroxyl radical (2OH^−^). These hydroxyl radicals are toxic to *M tuberculosis*. In fact, *M tuberculosis* is more susceptible to the redox products of Cu^2+^ than other bacteria such as *E. coli.* Within macrophages, the concentration of Cu^2+^ in certain phagosomes is high enough to be toxic for *M. avian*, a model for *M tuberculosis* [46]. Copper transport proteins (albumin, transcuprein) are essential for maintaining lower concentrations of copper. Transcuprein is a macroglobulin (270 KD) composed of several smaller proteins ranging from <100 to >100 KD. Transcuprein accepts Cu^2+^ from albumin so rapidly that it is considered the primary transporter [35]. Copper transport proteins are essential for maintaining lower concentrations of copper in surviving mycobacterial phagosomes. Mutant mycobacteria lacking this transport protein succumb to copper injury. Experimental animals (guinea pigs, mice) demonstrate a one-hundred- to one-thousand-fold decrease in isolates of *M tuberculosis* in lungs and lymph nodes when their outer membrane cannot eliminate the absorption of Cu^2+^ [45]. Even in the later stages of autophagy, hydroxyl radicals are lethal agents for surviving mycobacterial phagosomes [47].

## 6. Zinc (Zn^2+^) Deficiency in Malnourished Persons with Tuberculosis

Malnourished persons in East Africa (Ethiopia, Tanzania) often have a body mass index <18.5 kg/mg/m^2^ at the time of diagnosis of pulmonary tuberculosis [48,49]. The levels of Zn^2+^ are usually below the standard normal range. Adequate nutrition is associated with increased zinc storage, which is useful for macrophage activity against semi-dormant mycobacteria. In response, *M tuberculosis* has evolved an effective process for efflux of zinc from its internal environment. Since zinc cannot cross biologic membranes by passive diffusion, the energy for active transport is supplied by the action of cation transporting P-type ATPase [50]. Transmembrane proteins existing within the wall outside and inside of the mycobacterium assist in the efflux of Zn^2+^ from the pathogen. Zn^2+^ and Cu^2+^ combinations promoting mycobacterial morbidity/mortality have been thoroughly reviewed [51]. 

## 7. Iron (Fe^2+^) Deficiency and Excess May Participate in the Clinical Course of Pulmonary Tuberculosis

Malnutrition in the host is often accompanied by iron-deficiency anemia, which hinders the delivery of oxygen to the peripheral tissues. Lower levels of iron and ferritin, which are indictors of reduced reserves of circulating and stored iron, prior to the diagnosis of pulmonary tuberculosis are associated with treatment failure [52]. These deficiencies are more likely when sputum samples are found to be rapidly culture-positive for *M tuberculosis*. When a pulmonary macrophage is activated by interferon, phagosomes within the macrophage kill this pathogen. Hepcidin is a protein hormone that is synthesized in the liver and travels to the intestine to regulate iron absorption [53]. It also has a toxic effect on *M tuberculosis*. Elevated levels of either ferritin or hepcidin (iron overload or ferroptosis) have been observed in persons with an increased risk of tuberculosis infection [54]. On the other hand, a hepcidin deficiency is seen in persons with hepatitis C or with hemochromatosis. Another defense mechanism directed against *M tuberculosis* involves the generation of carbon monoxide (CO) through the action of heme oxygenase [55]. The result of carbon monoxide toxicity may be dormancy rather than death for *M tuberculosis*. Dormancy involves the expenditure of the smallest amount of energy possible through the elimination of functions like replication which are not immediately necessary [55]. Dormancy eliminates both the virulence of the pathogen and its detection by defense mechanisms of the host [12].

## 8. Magnesium (Mg^2+^)

Magnesium has a unique function in the RNA of *Mycobacterium tuberculosis.* Since magnesium (Mg^2+^) is abundantly available in the environment, it will readily combine with sections of RNA that are negatively charged. Structures within *M tuberculosis* RNA may be altered by conformational changes. These alterations may have consequences in the expression of genes. A portion of RNA (riboswitch) contains a section (M Box) which is conformationally stable. RNA is strongly electronegative. The binding of Mg^2+^ to the M box of *M tuberculosis* has consequences for mutational changes in protein structure, virulence, and accessibility to treatment [55]. Circular dichroism spectroscopy is helpful in imaging secondary structures before and after reactions (binding of ions, oxidative phosphorylation) as a way of understanding tertiary conformation. Circular dichroism uses far ultraviolet light (wavelength of 190–250 nm) along with near ultraviolet light (wavelength of 250–390 nm) to image sulfide bonds, amino acid residues, the orientation of bases, and peptide bonds. Divalent cations (Mg^2+^, Sr^2+^) can bind to the RNA of *M tuberculosis* and *B subtilis* [56]. When used for the analysis of divalent cation binding, it was found that there were no substantial differences between *B subtilis* bonds with Mg^2+^ and Sr^2+^. However, the binding of Mg^2+^ was favored over the binding of Sr^2+^ in the *M tuberculosis* M Box of the riboswitch [57]. Future studies may focus on the conformational changes resulting from the binding of divalent cations in the M Box to explain the effects of mutation upon virulence and antibiotic resistance within epidemics. 

## 9. Manganese (Mn^2+^) Is Present in Metalloproteins

Manganese-containing enzymes occur at the connection between the cycles of glycolysis and tricarboxylic acid (TCA). Phosphoenolpyruvate decarboxylase is responsible for decreasing three carbons to two at the end of glycolysis, at which point the critical products are either pyruvate or lactate. Acetyl CoA carboxylase is responsible for the production of citrate at the onset of the TCA cycle. A manganese-containing enzyme (arginase) occurs at the terminus of the urea cycle, at which point toxic ammonia is converted to ornithine plus urea [58]. Manganese has two pathways of attack on *M tuberculosis* surviving within pulmonary macrophages [59]. The first is to directly stimulate tumor necrosis factor (TNF)—related cascades. These pathway proteins include signal-related kinase (ERK) and signal-related c-Jun N-terminal kinase (JNK). These actions inhibit the survival of the pathogen in pulmonary macrophages. The second pathway to attack *M tuberculosis* is to stimulate TNF-related pathways by the phosphorylation of interferon genes [58]. Inhibitors of arginase are associated with improved blood flow to the heart/kidney in patients with vascular disease secondary to diabetes mellitus. For patients with diabetes, an area of tuberculosis research would be the efficiency of new arginase inhibitors [58].

## 10. Selenium Ions May Have a Direct Anti-*Mycobacterium tuberculosis* Effect

Levels of divalent cations such as zinc [50,51], iron [52,53], and selenium [60] are lower in study cohorts with tuberculosis than in matched groups without pulmonary tuberculosis. The correction of malnutrition in patients under treatment for tuberculosis will return levels of divalent cations to the normal range. However, the supplementation of antibiotic prescriptions with prescriptions for deficient cations has not been shown to improve outcomes. Among 23 trials involving 6482 study subjects, there was only a single study to indicate a possible improvement in clinical outcome by supplementing antibiotics with cation salts [61]. There is evidence to indicate that iron levels which are lower during infection may correct after successful antibiotic therapy by the release of the cation from storage sites and by the elimination of the blockade of intestinal absorption by hepcidin [62,63]. At some point during the treatment for pulmonary tuberculosis, it may become appropriate to restore selenium levels. Experience with selenium supplements revealed a narrow therapeutic window. As a result, the use of nanoparticles began [64]. This form of treatment alone [65,66] or in hybrid with an antibiotic [67,68] has generally met with success. 

## 11. Pulmonary Tuberculosis/COVID-19 Coinfection

The affinity for divalent cations may explain the symptoms of long COVID in survivors. Co-infection with other virus mechanisms may involve calcium movement through external channels or internal pathways. Analyses of case reports from areas where active tuberculosis is endemic have confirmed the hypothesis that the mortality rate must be higher for the co-infection than for either infection separately [69]. The complications of long COVID may include diminished memory/cognition, migraine headache, ischemic cerebrovascular attack with or without seizure, movement/musculoskeletal disorders with or without tremor, and a form of paralysis called Guillain–Barre syndrome [70]. A partial biochemical explanation for neurogenic stress/senescence may be the result of the affinity spectrum for Ca^2+^ and Cu^2+^ [71] which exists for certain amino acids (valine, glycine, lysine, leucine, phenylalanine). Calcium deposition salts (oxalate, phosphate) are thought to contribute to interference in critical points of electrical/mechanical signaling. This senescence mechanism may have positive side effects by inhibiting melanoma [72] and colorectal neoplasia [73] pathways in animal models. The affinity of calcium for structures performing the function of an anion, such as carbonate, citrate, chloride, oxalate, phosphate, sulfate, and urate, covers a wide range of molecular weights.

The outer layer envelopes of MERS corona viruses and SARS-CoV viruses [74] contain transmembrane channels for the movement of divalent calcium (Ca^2+^). In some studies, surface proteins take five-sided configurations (pentagons). An analysis of SARS-CoV E channel activity found the surface charges of proteins/lipids to be useful for the modulation of ion transit [75]. Calcium channel blockers may be protective in life-threatening adult respiratory syndrome infections during which intracellular calcium shifts into the cytoplasm from stores in the endoplasmic reticulum can be toxic to the bronchial epithelium/pulmonary alveoli [76]. The specific mechanism of benefit of calcium channel blockers (amlodipine, verapamil) in multi-drug-resistant pulmonary tuberculosis appears to be the shutting down of efflux channels by which the *M tuberculosis* rids itself of potentially lethal antibiotics [76]. In one study, COVID-19 patients treated with amlodipine for hypertension had a lower mortality rate than the controls not receiving a calcium channel blocker [77]. The mechanisms of clinical benefit of calcium channel blockers in viral infections may be like those in bacterial infections. These mechanisms would also involve calcium channels with ion pumps for movement against gradients [78]. Additional virus species which have been studied for voltage-dependent calcium channels potentially responsive to calcium channel blockers include *delta-corona* [79], *dengue* [80], *Ebola* [81], *hepatitis C* [82], *influenza A* [83,84], *rota* [85], and *West Nile* [86]. Interest in tuberculosis/influenza coinfection increases whenever an influenza epidemic occurs as well as when seasonal case numbers increase [86]. Studies have indicated that patients with pulmonary tuberculosis may be more susceptible to invasive influenza infection. In addition, episodes of influenza infection may be associated with a lack of immune protection from invasive pulmonary tuberculosis [87,88]. A mouse study has demonstrated a 100% mortality with coinfection of influenza virus and tuberculosis, as opposed to a 5% mortality with influenza virus infection alone [88]. Certain bacteria (*Cholera vibrio*, *M tuberculosis*) and viruses (*Ebola*, *Dengue*, *influenza*) have been shown to utilize host endo-lysosomal cation channels for the transport of nutrients [89]. 

## 12. Calcium (CA^2+^) Mechanisms in Persons with Diabetes Mellitus as a Risk Factor for Tuberculosis and Parasitic Disorders

There is no convincing evidence that persons with pulmonary tuberculosis have a greater risk of new-onset diabetes mellitus than other forms of pneumonia. Persons with diabetes mellitus experience an increased risk of complications of invasive tuberculosis than persons without diabetes mellitus. The increased risk of complications of pulmonary tuberculosis experienced by persons with diabetes mellitus may be explained in part by the decreased activity of macrophages, monocytes, and T-lymphocytes [90].

Relatively lower calcium and vitamin D levels have been described in a review of patients with type 2 diabetes mellitus [91]. The ratio of calcium to magnesium is higher in persons with diabetes mellitus than without diabetes. Therefore, the level of magnesium is also lower in the study group with diabetes mellitus. In addition to levels of serum calcium and serum magnesium being lower than normal in persons with diabetes mellitus, the level of magnesium is inversely related to the degree of pulmonary injury in persons with tuberculosis [10].

As part of the immunologic phase of beta cell destruction in type 1 diabetes, a higher intracellular calcium concentration is noted. Excessive intracellular calcium concentrations cause apoptosis in the Beta cells of the Islets of Langerhans. Thioredoxin-interacting protein, associated with Beta cell dysfunction, may be reversed with glucagon-like peptide receptor agonism and Ca^2+^ channel blockade [91,92,93,94,95]. Vitamin D and calcium (Ca^2+^) are associated with multiple reactions culminating in the secretion of insulin [96]. Several of these reactions include the promotion of the expression of the insulin gene [90] and the organization of the binding of Ca^2+^ in the cytoplasm of Beta cells by the protein calbindin [96] along with the regulation of oscillations of Ca^2+^ in the cytoplasm of Beta cells, by which the secretion of insulin occurs in a pulsatile rhythm. The activation of endopeptidase promotes exocytosis of insulin from Beta cells [97]. Intravenous insulin infusion in a pulsatile rhythm was found to be superior to continuous infusions with intermittent injections, causing a reduction in glycohemoglobin A1c from levels greater than 8% down to 7.0%, along with the elimination of serious low glucose events [98]. This was also true for the preservation of kidney function in a longitudinal study of individuals with type 1 diabetes with albuminuria [99]. 

Ion channels allow calcium, sodium, and potassium to pass through cell membranes according to electrical gradients which are strengthened every time the process is repeated. Since calcium is central to neuromuscular function, calcium channels are useful to Schistosomes (haematobium, japonicum, mansoni) in their fresh water free swimming phases (miracidia, cercariae). Praziquantel is the leading agent against Schistosomes. One of the targets of praziquantel is the family of ion channels known as transient receptor potentials (TRP) [100]. In mammals, TRP channel A1 interacts with sensory (light, chemical), pain (nociceptive), and inflammation cascade signals. Some evidence indicates a stimulation of the host (mammal) TRP by praziquantel as an additional mode of defense. The intracellular message of TRP may be transmitted according to the concentration of Ca^2+^. Wide-ranging research indicates TRP interaction with cell functions like autophagy/apoptosis [12], tumor initiation/progression [101], and inflammation/nociception [102].

## 13. Chelation Therapy

Chelation therapy for the reduction of toxic levels of divalent cations has become a subject of extensive study (Table 4). A systematic review and meta-analysis of the prevention of cardiovascular complications with chelation therapy [103] involved the use of ethylenediamine tetra acetic acid (EDTA) = C_10_H_16_N_2_O_8_. This review/meta-analysis emphasized studies of individuals with diabetes mellitus who achieved clinical improvement goals through the application of chelation protocols [101]. While most of the chelation projects report a removal of calcium, several focus specifically upon lead [104,105,106] or cadmium [107,108,109]. Additional studies present results with chelation of the entire list of divalent cations [110,111,112,113,114,115], which might be present in the air or water or soil as environmental contaminants. 

## 14. Discussion of New Concepts for Development of Medications against Pulmonary Tuberculosis

A new approach involving enzymes of *M tuberculosis* uses three-dimensional imaging to determine the site for the attachment of cadmium (atomic weight 112 g). The use of one of the heavier divalent cations enables the demonstration of the unfolding (denaturation) of the protein enzyme [116]. Other novel uses of three-dimensional molecular imaging may identify sites that can be targeted to interfere with critical *M tuberculosis* survival and pathogenicity.

The success of the bacillus pathogen *M tuberculosis* in surviving the defense mechanisms of the host has been attributed to its ability to remain alive in a relatively dormant state after being ingested by pulmonary macrophages (Table 5). Given this protected environment, antibiotics and toxins may be unable to directly confront the invader. One technical problem that inhibits research is the slow growth of *M smegmatis*, which is safer for individuals working in the laboratory [117]. Genomic adaptations to the intake and efflux of divalent cations for their nutritional values vs. their cellular toxicities are being reported in remarkable detail [118].

Another instance where remarkable detail has emerged involves the secretion of a tuberculosis necrotizing toxin, which can kill a macrophage infected with *M tuberculosis* [117]. Five required reactions control the opening of exit passages for the toxin to leave a particular member of *M tuberculosis*. When the necrotizing toxin reaches the cytosol of the macrophage, the work of killing the macrophage proceeds by attacks on its walls. This process is called permeabilization [118,119]. 

Current research highlights the tug-of-war between the pathogen and the host in the acquisition of an adequate number of divalent cations for survival within a macrophage as opposed to the flooding of the macrophage cytosol with excess amounts of divalent cations, such as calcium/copper/zinc, that will serve as toxins [115]. The failure of the human host defense mechanisms to detect a dormant state of *M tuberculosis* [9] allows this pathogen sufficient time to create nodular granulomata [120]. A transthoracic needle biopsy of these solitary nodes will reveal live mycobacteria [121]. 

A channel for Ca^2+^, Na^1+^, and K^1+^ (peizo1) on the surface of the red blood cell (RBC) controls the shape and the surface contour of the RBC when stimulated mechanically [122]. A frequent form of stimulation is stretching. The concave shape can become flat while the smooth surface can become spindly, like that of a hedgehog or a sea urchin. The process of the development of surface spines is called echinocytosis. Certain parts of Africa, where there is the problem of red cell hemolysis due to the assumption of the shape of a sickle, have reported protection from infestation with *P falciparum*. The protection occurs with the sickle trait heterozygous state (SA) but not with the homozygous state (SS) [123]. Certain parts of Africa may also experience protection from infestation by *P falciparum* if there is a mutation in the gene for the peizo1 channel [124]. Alterations in shape and surface contours present a barrier to attachment and adherence of the merozoite phase of *P falciparum*. While the entry of merozoites is blocked by this mutation in the gene for expression of the peizo1 ion channel, there is no impediment to egress [122]. Concentrations of the cations (Ca^2+^, Na^1+^, K^1+^) are also changed following the mechanical stimulation of the piezo1 channel. Intracellular levels of Ca^2+^ are increased, which generally favors the process of egress of the merozoite. Dehydration was suggested by an elevation of sodium greater than potassium [121]. A unique experimental mouse model quickly succumbed to cerebral malaria following invasion with Plasmodium species. The use of new pharmacological agents which activated the gene for expression of the peizo1 gene demonstrated a dramatic protection from invasive Plasmodium species [122]. The mechanism appeared to be the inhibition of adhesion to the outer surface of the lipid bilayer. 

Three-dimensional imaging techniques identify the molecular physiochemical structural sites of enzyme proteins. This permits the design of new antibiotics combining efficiency with reduced side effects. Divalent cations may be useful in antimicrobials by binding to reactive sites. In addition to the design of new antibiotics, calcium channel blockers may represent important adjuncts by preventing the efflux of bactericides from the pathogen, permitting a lower dosage and fewer side effects. 

One of the more difficult future research topics will be the development of testing to detect and/or attack dormant pathogens within macrophages before they reach permanent safety inside of a nodule. Chagas disease cardiomyopathy caused by *T cruzi* infestation is potentially fatal. By means of CRISPR techniques, work with pluripotent stem cell-derived cardiomyocytes has developed the capacity to track the intracellular movement of calcium. This technique, involving calcium indicator proteins, may be applied to other intracellular pathological processes in which functions such as rhythmical contraction can be analyzed. The CRISPR–Cas9 system enables the imaging of conformational changes during the activity of enzymes using magnesium ions. Two sites on the DNA must be cut simultaneously to ensure new DNA for a desired endpoint (mutation, stem cell conversion). Dr. Hong Li and associates at Florida State University show how magnesium ions (Mg^2+^) are imaged as a chemical bond is broken through the CRISPR–Cas9 technique [56].

In some areas, the twindemic of tuberculosis and malaria may attack a population at the same time. Recently, malaria has returned to the United State of America after having been proclaimed eradicated 75 years ago. Although global warming has often been blamed, an editorial in *The Lancet* suggested that urban growth, international rapid transit, and drug resistance were as important as climate change [125,126]. Medication resistance becomes more likely when the host is attacked by different pathogens at the same time [127] and when multiple antimicrobials are prescribed in persons immunosuppressed by malnutrition [128,129,130]. Of particular concern is the capacity of pathogens (*M tuberculosis*, *P falciparum*) to enhance drug resistance by utilizing calcium channels for drug extrusion [131]. Calcium channel blocking agents (verapamil, diltiazem, amlodipine) may be useful as inhibitors of drug extrusion by pathogens and require additional study (Table 5) [7,76,77,131,132]. Possible parallels for future treatment development may be associated with recent studies of transmembrane conductance. Single cell parasites (*E histolytica*, *T gondii*, *L donovani*, *T cruzi*, *P falciparum*) are the target of recent studies of transmembrane intake conductance of nutrients and excretion conductance of waste. One approach is the patch-clamp technique, which quantitates the transit of ions (Na^1+^, K^1+^, Ca^2+^, Mg^2+^) through lipid bilayers that would otherwise be impermeable [132,133]. The protein and lipid components of transit channels organize the gradients of membrane potentials [132]. Ions move through the channels according to their concentration gradients and/or their electrochemical gradients. 

## Figures and Tables

**Figure 1 ijms-25-09775-f001:**
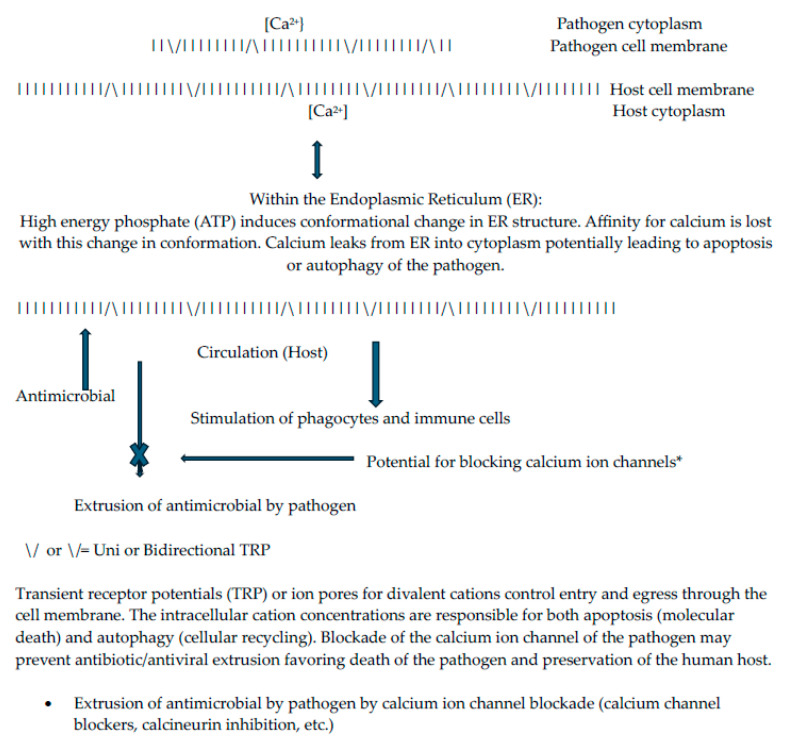
Transient receptor potential (TRP) for divalent cations (shown for Ca^2+^).

**Figure 2 ijms-25-09775-f002:**
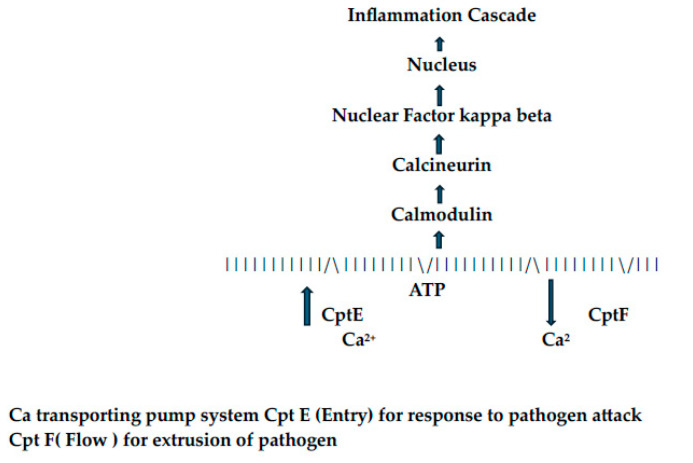
Calcium transporting pump system: initiation of host inflammatory cascade following attack by pathogen.

**Table 1 ijms-25-09775-t001:** a. Molecular actions of calcium (Ca^2+^). b. Toxicity of pathological concentrations of divalent cations.

**a**						
**Blood Coagulation**						
	Conformational changes allow prothrombin to bind efficiently to phospholipid surfaces					
	Promotes platelet adhesion to blood vessel endothelium with von Willebrand factor					
**Bone Cortex**						
	With phosphate increases mass during growth phase					
	With phosphate and exercise increases strength					
**Cell Signaling**						
	Stimulates mitochondrial oxidation of ketoglutarate					
	Stimulates mitochondrial oxidation of pyruvate dehydrogenase					
**Digestive System**						
	Stimulates gastric acid secretion					
	by Vitamin D-activated calbindin, contributes to intestinal absorption of calcium					
**Kidney**						
	Reabsorbed passively by proximal tubule					
	Ca^2+^ sensing receptor controls absorption in loop of Henle					
	Klotho gene controls calcium absorption with transient receptor protein					
**Muscle**						
	Contributes to orderly release of calcium from sarcoplasmic reticulum					
	Contributes to orderly return of calcium to sarcoplasmic reticulum					
	Increases expression of ryanodine receptor involved in Ca^2+^ release					
	Activates/deactivates actin–myosin for contraction/relaxation					
**b**						
**Element**	**Ion**	**Neuromuscular**	**Cardiovascular**	**Gastrointestinal**	**Renal**	**Other**
**Calcium**	Ca^2+^	+	−	+	−	lung
**Cadmium**	Cd^2+^	+	+	+	−	skin
**Copper**	Cu^2+^	+	+	+	+	-
**Iron**	Fe^2+^	−	−	+	+	lymphatic
**Lead**	Pb^2+^	+	−	+	+	skin
**Manganese**	Mn^2+^	+	+	+	−	-
**Magnesium**	Mg^2+^	+	+	−	−	reproductive
**Selenium**	Se^2+^	+	+	+	+	skin
**Silver**	Ag^2+^	−	−	−	−	skin
**Zinc**	Zn^2+^	+	−	+	+	reproductive

**Table 2 ijms-25-09775-t002:** Effects of calcium-related hormones upon serum calcium concentration.

Parathyroid Hormone			
	Organ level		
		Stimulates kidney to synthesize vitamin D Increases calcium absorption in kidney tubule Inhibits phosphorus absorption in kidney tubule Decreases calcium phosphate bone mineral mass Cooperates with vitamin D in increasing bone mass	
	Cell level		
		Increases expression of alkaline phosphatase Increases expression of bone morphogenetic protein Increases expression of collagen type1 alpha Increases expression of osteoblast transcription factor (Tmem119) Increases expression of calcium-binding protein (osteocalcin)	
Vitamin D *			
	Organ Level		
		Increases intestinal absorption of calcium Decreases cytokines of inflammation	
	Cell level		
		Inhibits inflammation cascade at nuclear factor kappa beta Decreases cytokines of inflammation Supports functions of macrophages Activates nitric acid synthase in endothelial cells Decreases expression of receptor for advanced glycolated end-products	
Calcitonin			
	Organ level		
		Bone	
			Contracts osteoclasts Diminishes osteoclast mobility Decreases loss of bone mineral mass
		Kidney	
			Decreases reabsorption of calcium, magnesium Decreases reabsorption of phosphate Decreases reabsorption of sodium => diuresis
	Cell level		
		Binds to its receptors on osteoclasts Promotes vitamin D production enzymes Adenyl cyclase for regulation of cell growth by cyclic AMP (second messenger) Inosityl triphosphatase for interaction with calcium/calmodulin/calcineurin cascade	

* Vitamin D meets the criteria for definition of a hormone.

**Table 3 ijms-25-09775-t003:** Calcium metabolism in pulmonary tuberculosis.

A. Vitamin D increases serum calcium level due to:
1. Increased sunlight exposure with increased synthesis of vitamin D by skin
2. Increased 1-alpha hydroxylase from lung, intestine in addition to kidney
3. Overheating with dehydration
B. Parathyroid hormone increases serum calcium level due to:
1. Lysis of bone cortex
2. Promotion of synthesis of vitamin D
3. Increased expression with hyperphosphatemia of kidney failure
C. Calcitonin regulates increased serum calcium levels due to
1. Inhibition of bone cortex lysis by parathyroid hormone
2. Increased expression during inflammation which might injure kidney function

**Table 4 ijms-25-09775-t004:** Reported benefits from calcium chelation therapy with EDTA.

Pathology	Results
**Diabetes Mellitus**: decreased insulin secretion	improved glucose control
**Cardiovascular Disease:**	
Heart: angina pectoris, use of nitroglycerine	fewer events, decreased use of nitroglycerine
Vascular	
Central: dizziness/vertigo	fewer events
Peripheral: ulcers, gangrene	healing, no amputations

**Table 5 ijms-25-09775-t005:** Life cycle of *M Tuberculosis* within the human host after exposure.

Stage of Disease Activity	Clinical Evidence	Treatment
Primary infection: dormant or latent TB infection	Pulmonary nodules with hibernating pathogen (autophagy completed). Few symptoms, evidenced by skin testing and chest X-ray only	Antibiotics per protocol *
Active but quiescent or evasive	One-fourth to one-half within the macrophages (autophagy commenced). Symptoms primarily systemic (fatigue, fever, loss of appetite and weight, weakness)	Antibiotics per protocol *
Active: aggressive pulmonary	Pulmonary cavitation with respiratory symptoms added to above	Antibiotics per protocol *
Active and disseminated	Miliary: multiorgan involvement	Multiple antibiotics *

* Potential use of calcium ion blockade requires additional investigation.

## Data Availability

Not applicable.

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
