# Peer review of "Role of Divalent Cations in Infections in Host–Pathogen Interaction"

_ijms, 2024, doi:10.3390/ijms25189775_

Round 1

Reviewer 1 Report

Comments and Suggestions for Authors

The manuscript by D’Elia and Weinrauch describes the role of the divalent cations in host-pathogen interaction. While the subject is incredibly interesting and the authors as MDs provide fresh insight for the molecular biologist there are numerous problems with the manuscript that need to be addressed before it is publishable.

Major issues:

-          The introduction is completely missing. I would expect to find there general information about the importance of divalent cations for cell function as well as some examples of host-pathogen interactions, as there are many types of that.

-          Chapter 4 “survival mechanism” – while reading this paragraph I was confused on what kills what. Several reads are required to understand this paragraph I suggest to streamline the text and also provide a Figure with explanation.

-          As mentioned above lack of Figure that explain the biological process is really concerning. I would expect at least several Figures to help the readers understand more complex parts of the review.

-          There is an issue with references starting with chapter 5. The references 34,35 are lost (they are correctly used just above in previous paragraph. Later references re sometimes correct and sometimes missing. For example, Chapter 9 reference [51] should be [50]. Please check all the references numbers.

-          Table 4 is full of errors. Calcium citrate formula is completely wrong. It should be Ca3(C6H5O7)2. Calcium urate formula should be corrected to put the stoichiometry in the subscript. Also the formula should have Ca at the beginning, not middle.

-          Chapter 11 – I don’t understand the part about carbonyl bond involvement in the binding. The carbonyl bond in the mentioned amino acids is present only in the peptide bond, so how the size of aliphatic or aromatic amino acid residues are important? Please explain

-          Only 27 references out of 108 are from the last 5 years. While this is not necessary an issue I see that some recent concepts are missing for example ferroptosis (reviewed here https://pubmed.ncbi.nlm.nih.gov/38917759/). Please add some more recent papers.

Medium severity issues:

-          The title is a bit misleading as the sole focus of mycobacterium, not different pathogens.

-          Table 1 – First formula should be changed to ion formula. Second, I don’t think that atomic mass is important for the readers, but rather I would put here the major function of particular metal ion for cells/organism.

-          Page 2 – second to last line. “bone is an important target goal” – while true this is a great simplification. Please provide more molecular biology details.

-          Page 5 line 7 – please provide the name of the transporter

-          Chapter 8 – I suggest to elaborate on molecular basis of magnesium ion binding to M-box causing mutations.

Editorial issues:

>        ORCID of the 2nd author is invalid

>        The corresponding author is not given

>        Abstract line 3 “effect” shouldn’t there be affected?

>        I’m not sure, what is the purpose of enumeration in the abstract?

>        Throughout the text there are multiple examples of gaps between words. E.g. line 12 of introduction, page 2 line 37 (resolution) and many more.

>        The currently accepted notation of charges is Ca2+ not Ca++, please correct for all metal ions through the text

>        Introduction line 15. Naming calmodulin as calcium message transporter is a bit vague biologically. Please correct.

>        Page 4 M. tuberculosis and E. coli should be in italic.

>        Chapter 8 and 9 title – the ion is missing in the title (Mg2+, Mn2+), which is present in titles before.

>        Chapter 11 – there is a sentence with all first capital letters, where it is not necessary

>        Page 8 – EDTA descriptive formula is not needed. Please remove. Also put the stoichiometry numbers of the EDTA formula in subscript.

>        Table 5 – Vascular should be in bold

>        Funding, data availability and conflict of interest is not written.

Author Response

To the reviewers:

Thank you for your kind and constructive reviews. We have endeavored to respond to each comment by modifying the text in to improve clarity and hope that the revision meets all the Editorial needs. We also hope that this manuscript will stimulate more research in this area. There are some comments however that we must note:

  1. The ORCID number of the second author is, in fact, correct
  2. The purpose of the enumeration in the abstract is referencing
  3. The spacing that has been a problem resulted from the switch from Calibri to Palatino typescripts and margin changes and we have endeavored to fix this. Our apologies!
  4. The formula in Table 4 for calcium urate was actually correct according to the NIH (Pubchem), however we have removed this table from the manuscript as per your suggestion
  5. Chapter 4 has been rewritten and the errant references have been placed in correct order
  6. We have removed the tables that may have caused confusion and avoided comments related to carbonyl bonding that are confusing and not necessary to this presentation
  7. The addition of a longer introduction has been made. As clinicians we often forget that our laboratory results are not just something to be aware of and adjust but are rather the result of subcellular and cellular processes . Our apologies for this misstep vis a vis the readership of the Journal.
  8. We have modified and added tables relating to the function of calcium in the human and the toxicity if divalent cations at excess concentrations which we hope will clarify the concepts to the reader
  9. We have broadened the focus regarding organisms discussed 
  10. We have changed to ionic formulae as requested and made the editorial suggestions that you have identified
  11. As we have expanded the manuscript in response to requests for additional information, the bibliography has also expanded to include more recent papers among the 19 additional articles cited.
  12. I believe that we have responded tall of your critiques. If you feel that a better title would serve we have no objections. 

We hope that the manuscript reads better and is now acceptable. Please accept our apologies for any problems with lining up the tables and their placement. We are not particularly adept at this aspect of publication.

Thank you,

Larry Weinrauch MD

Reviewer 2 Report

Comments and Suggestions for Authors

The current review article titled "Role of Divalent Cations in Infections in Host-Pathogen Inter-action" provides an overview of the involvement of divalent cations in host-pathogen interactions, particularly focusing on their role in infections. The article is well-researched and covers a wide range of topics related to the biochemical and molecular mechanisms by which divalent cations, such as calcium and magnesium, influence the survival and virulence of pathogens like Mycobacterium tuberculosis within host cells. The article covers a broad spectrum of divalent cations and their roles in various cellular processes, providing a detailed account of the molecular mechanisms involved in pathogen-host interactions. Here are some suggestions for improvement:

1.      The article includes basic scientific information, such as the molecular weights of elements (table 1 and table 3), which is more suited for an introductory textbook or general science communication rather than a scholarly review. This detracts from the focus and depth of the review. Instead, the focus should be on the functional and mechanistic roles of these cations in specific host-pathogen interactions.

2.      The article's title suggests a broad exploration of the role of divalent cations across various host-pathogen interactions. However, the examples provided are limited to a few pathogens, primarily Mycobacterium tuberculosis, which might not justify the generality implied by the title. Either broaden the scope by including additional pathogens that are influenced by divalent cations or modify the title to reflect the narrower focus of the review. Including examples from other bacterial, viral, or fungal pathogens would enhance the article's applicability and appeal to a broader audience.

3.      Current Issue: The article emphasizes the molecular mechanisms underlying pathogen-host interactions in the introduction, but the discussion in the main body is not consistently deep in this regard. Specifically, the review could delve more into how divalent cations interact with various signaling pathways that influence infection outcomes. To better fit the scope of a molecular sciences journal, the review should integrate more mechanistic insights, possibly by including recent findings from studies using advanced molecular techniques such as CRISPR, RNA-seq, or proteomics. If the current level of detail is maintained, the article might be more suitable for a broader, less specialized journal.

The article presents valuable information on the role of divalent cations in infections, but it would benefit from a more focused and detailed discussion on molecular mechanisms, a broader scope of pathogens, and the removal of basic scientific information that is not central to the article's theme. Addressing these points would significantly enhance the article's scholarly impact and relevance to the target audience.

Author Response

To the reviewers:

Thank you for your kind and constructive reviews. We have endeavored to respond to each comment by modifying the text in to improve clarity and hope that the revision meets all the Editorial needs. We also hope that this manuscript will stimulate more research in this area. There are some comments however that we must note:

  1. The ORCID number of the second author is, in fact, correct
  2. The purpose of the enumeration in the abstract is referencing
  3. The spacing that has been a problem resulted from the switch from Calibri to Palatino typescripts and margin changes and we have endeavored to fix this. Our apologies!
  4. The formula in Table 4 for calcium urate was correct according to the NIH (Pubchem), however we have removed this table from the manuscript
  5. We have removed the tables that may have caused confusion and avoided comments related to carbonyl bonding that are confusing and not necessary to this presentation
  6. The addition of a longer introduction has been made. As clinicians we often forget that our laboratory results are not just something to be aware of and adjust but are rather the result of subcellular and cellular processes . Our apologies for this misstep vis a vis the readership of the Journal.
  7. We have remove the molecular weights and referred to role played by divalent cations accross various host pathogen interactions
  8. We have modified and added tables relating to the function of calcium in the human and the toxicity if divalent cations at excess concentrations which we hope will clarify the concepts to the reader
  9. As we have expanded and broadened the manuscript in response to requests for additional information, the bibliography has also expanded to include more recent papers.
  10. We would have no objection to a change in title that would better reflect Divalent Cations in Infections in Host-Pathogen Interaction and await your further thoughts

We hope that the manuscript reads better and is now acceptable. Please accept our apologies for any problems with lining up the tables and their placement. We are not particularly adept at this aspect of publication.

Thank you,

Larry Weinrauch MD

Round 2

Reviewer 1 Report

Comments and Suggestions for Authors

I appreciate the changes made by the authors. The manuscript fells much more evolved and it’s a pleasure to read. However there are several editorial issues that must be corrected before publication:

- The ORCID of the second author is unfortunately incorrect as it is missing 2 numbers (at least in the PDF version I see). The correct number I presume it is 0000-0003-1357-9528

- there is a repeated fragment about the circular dichroism. The same text is written on Page 9 (top of page) and on Page 13. Please choose where the paragraph is more appropriate. I believe it more suits the subject on Page 9 than 13.

- The references has been corrected, however I still detect a missing refence. On Page 7 in the start of the 5th paragraph. There are sentences about copper homeostatis. The reference 41 is cited, however the same reference is used (correctly) in the sentence above to described finding on procalcitonin. Reference 42 is correctly cited later in the paragraph, therefore I believe that reference for basic information on copper is just missing.

-  The last paragraph “Discussion of New Concepts for Development of Medications Against Pulmonary Tuberculosis”, however I’m not sure why a switch was made from M. tuberculosis to P. falciparum etc? Is it to draw some possible parallels between treatment development?

Reviewer 2 Report

Comments and Suggestions for Authors

The article has greatly improved thanks to the author's modifications and improvements, addressing most of the concerns.

Author Response

Thanks for your comments, we appreciate it.